# VEGA: Visual Expression Guidance for Referring Expression Segmentation

## Abstract

Referring expression segmentation aims to segment a target object described by a given linguistic expression in an image. Unlike the unimodal segmentation taking predefined categories, this task takes the free-form linguistic expression that contains a single attribute or more than one attribute (*e.g.*, location, color and action) related to the target object. However, the given linguistic information is only some part of information on the target object. In contrast, the image contains more additional information for the target object, including the unique information that is hard to describe in linguistic expression. Motivated by this, we propose a novel **V**isual **E**xpression **G**uid**A**nce framework for referring expression segmentation, **VEGA**, which enables the network to refer to the visual expression that complements the linguistic expression information to improve the guidance capability. Since the image includes information related to both target and non-target regions, it needs to meticulously identify and selectively extract the useful information relevant to the target object. Therefore, we introduce a novel visual information selection module that flexibly selects the semantic visual information related to the target object to produce the visual expression, enhancing the adaptability to diverse linguistic and image contexts for robust segmentation. Furthermore, the proposed module allows each token of the visual expression to consider the visual contextual information by exploiting the global-local linguistic cues, thereby enhancing the capacity to understand the context of the target region. Our method consistently shows strong performance on three public benchmarks for referring expression segmentation, where it surpasses the existing state-of-the-art methods.

## 1 Introduction

Referring expression segmentation (RES) (Hu et al., 2016; Liu et al., 2017; 2023b; Hu et al., 2023) is one of the challenging vision-language tasks, and can be applied in various applications such as human-robot interaction and the object retrieval. Given an image and a natural language expression describing a target object in the image, RES aims to segment the specific object region guided by the language expression.

Unlike the single modal segmentation (Long et al., 2015; Cao et al., 2022; Heidari et al., 2023; Shim et al., 2023; Xie et al., 2021) based on fixed categories, the RES addresses the free-form language expression that is ambiguous and complex. For instance, the language expression can be given as a word that represents a single attribute (*e.g.*, location), such as *"left"*, or as a phrase or sentence that represents more than one attribute (*e.g.*, color and relation), such as *"pink shirts on the sofa"*. However, the given language expression contains *only some part* of the information about the target object. In contrast, as shown in Figure 1, the image contains more various information for the target object than the language expression, such as the location, color, size and relationships between objects and the unique information that is difficult to describe in the language expression. Since the image contains the information related to both target and non-target regions, it needs to meticulously identify and selectively extract the appropriate information relevant to the target object. Therefore, we focus on selecting the useful visual information to produce a *visual expression* that can guide to the target object, improving the guidance capability by complementing the linguistic expression information. In this paper, a set of the visual and linguistic expressions is called a *guidance set*.

The existing studies (Feng et al., 2021; Yang et al., 2022; Wang et al., 2022) have focused on the multi-modal fusion approach, which enables vision features to effectively refer to the language ex-

Figure 1: Conceptual diagram of the vision information with language expressions. The visual contexts encompass more diverse information than the linguistic expression.

pression information. Some other studies (Ding et al., 2022; Tang et al., 2023) allow language features to refer to the vision information by employing the language-vision cross-attention mechanism to improve the comprehension for the linguistic expression. Although this approach enables the network to refer to the enhanced linguistic information, there is a limitation in that the elements of the guidance set are only the language tokens in the existing methods. Therefore, these method cannot provide the unique visual information sufficiently, which helps with robust segmentation of the target region. To produce the visual expression for the robust segmentation, we explore how to flexibly select the visual semantic information relevant to the target object.

In this paper, we propose a novel *visual expression guidance* framework that enables to refer to the visual guidance expression with the linguistic guidance expression to take full advantage of both the visual knowledge and the linguistic knowledge. The proposed framework is distinct from the previous work in that we produce the visual expression to enhance the robustness of the guidance set. The visual expression complements the given linguistic information by providing the unique information of the visual features relevant to the target region. To better produce the visual expression, we design a novel visual information selection module that combines the flexible selection with an effective utilization of global-local contextual cues (*i.e.*, sentence level and word level contexts). The proposed module flexibly selects the visual semantic information via the top-$k$ candidate pixel selection to strengthen the adaptability to diverse linguistic expressions and image contexts, leading to the robust segmentation of the accurate regions. Furthermore, by considering both the comprehensive and specific attribute contexts, the visual expression can contain richer visual information on the target object. To further improve the ability to understand the context of the target region, each token of the visual expression acquires the visual context information by considering the relationship between each token. In this way, our method builds up a novel **V**isual **E**xpression **G**uid**A**nce framework for referring expression segmentation.

We demonstrate the effectiveness of the proposed framework on three public benchmark datasets for the referring expression segmentation. Extensive experiments and ablations show that our visual expression generation framework enable to enhance the guidance capability for the target regions. In particular, our approach outperforms the previous state-of-the-art methods on RefCOCO, RefCOCO+ and G-Ref datasets. Our contributions are summarized as follows:

- We propose a novel visual expression guidance framework for referring expression segmentation, **VEGA**, which enables the useful visual knowledge to be used as a guidance set element with the linguistic knowledge to complement the given linguistic information and provide the unique visual information. This framework represents a new approach to enhance the guidance set capacity in referring expression segmentation.

- We introduce a novel visual information selection module that flexibly selects the visual semantic information relevant to the target object to produce the visual expression, enhancing the adaptability to various linguistic expressions and image contexts for the robust segmentation.

- The proposed module effectively harnesses the global-local linguistic cues to enables each token of the visual expression to consider the visual contextual information, improving the ability to comprehend the context of the target region.

- Our VEGA consistently demonstrates strong performance and surpasses the current state-of-the-art methods on three public referring expression segmentation benchmarks.

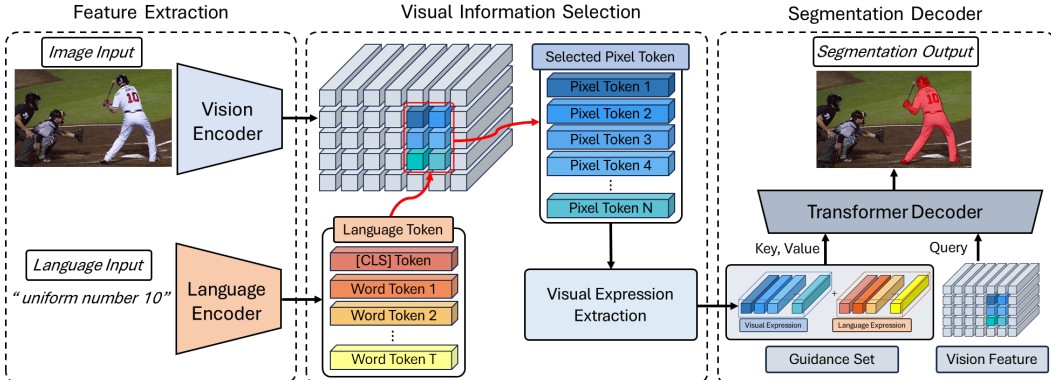

Figure 2: Overview of the proposed framework, VEGA. Our method produces the visual expression to improve the robustness of the guidance set capacity.

## 2 RELATED WORK

**Referring Expression Segmentation.** Different from the conventional segmentation (Ronneberger et al., 2015; Long et al., 2015; Cheng et al., 2022; Heidari et al., 2023) based on predefined categories, referring image segmentation aims to find the target object according to the unrestricted language expression. Hu et al. (2016) first proposed the referring expression segmentation network that concatenates the language features and vision features to fuse them. Recent researches (Kim et al., 2022; Wang et al., 2022; Yang et al., 2022) have explored on the better multi-modal fusion for this task. ReSTR (Kim et al., 2022) and CRIS (Wang et al., 2022) fused the vision and language features after the feature extraction. LAVT (Yang et al., 2022) showed great advance by performing the early fusion in a Swin Transformer (Liu et al., 2021). Following this work, several studies (Yang et al., 2023; Liu et al., 2023a; Tang et al., 2023) adopted the early fusion approach for the effective fusion. Another recent study has focused on improving the comprehension for the language expression. VLT (Ding et al., 2022) enhanced the understanding ability to the language expression by referring to the vision features through the cross-attention layers. More recent researches (Wu et al., 2022; Liu et al., 2023a) have focused on improving the diversification of the language expression samples, such as the negative sample augmentation. Unlike these approaches, we focus on producing the *visual expression*, which can provide the unique visual information sufficiently, to enhance the guidance set capacity for the robust segmentation of the target regions.

**Token Selection.** Recent studies have explored the token selection in various tasks. TS-ViT (Zhou et al., 2022) proposed a drop-in token selection method to improve the selectivity of the self-attention and enhance the robustness of the transformer models on image classification, semantic segmentation and NLP tasks. For patch selection in large images, Cordonnier et al. (2021) proposed the top-$k$ method to aggregate information from the different patches in a flexible manner. For video object segmentation, Seong et al. (2021) proposed a top-$k$ guided memory matching method, resulting in efficient and robust fine-scale memory matching. Cheng et al. (2021) also introduced a new top-$k$ filtering scheme for the attention-based memory read operation. TS2-Net (Liu et al., 2022) proposed a token shift and selection transformer that adjusts the token sequence dynamically and selects informative tokens in both temporal and spatial dimensions from videos on text-to-video retrieval.

We propose a visual information selection module that combines the top-$k$ pixel selection with the utilization of global-local linguistic cues to flexibly select the useful visual information associated with the target region and effectively capture the visual contexts, thereby improving the ability to comprehend the context of the target region.

## 3 METHOD

We propose a novel visual expression guidance framework on referring expression segmentation, **VEGA**, which leverages the visual expression as a element of the guidance set with the linguistic expression to enrich the semantic information that robustly guides the network to the target region. The overall architecture is illustrated in Figure 2. We first describe the vision and the language

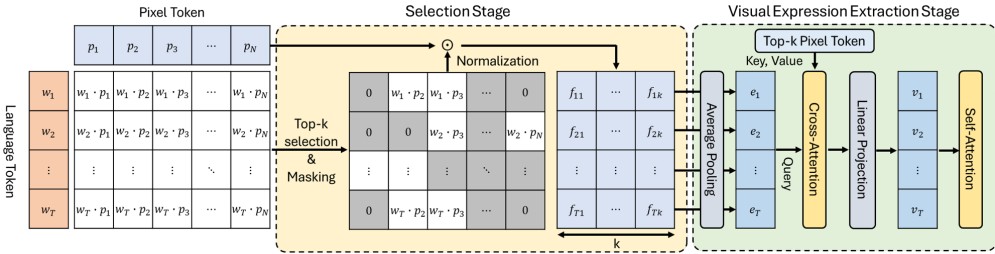

Figure 3: Detailed illustration of our visual information selection module, consisting of the selection stage and the visual expression extraction stage. This module flexibly collects the semantic information relevant to the target regions, and considers the contextual information of the visual expression to improve the capability to understand the context of the target region.

feature extraction in Section 3.1, and then introduce the visual information selection module that flexibly selects the semantic information to produce the visual expression in Section 3.2. Finally, we explain the segmentation decoder referring to the guidance set that contains visual and linguistic expressions in Section 3.3.

## 3.1 VISION AND LANGUAGE FEATURE EXTRACTION

Given the input image $\mathcal{I}$ and the referring language expression $\mathcal{Q}$ that consists of $T$ words, the vision encoder extracts the vision features $F_i \in \mathbb{R}^{H_i W_i \times C_i}$ at each stage $i \in \{1, 2, 3, 4\}$ and the language encoder extracts the linguistic expression tokens $Q_l = [w_1, w_2, ..., w_T] \in \mathbb{R}^{T \times D}$. Note that $H_i$, $W_i$, $C_i$ and $D$ denote the height, width, channel dimension of the feature maps at the $i^{th}$ vision stage, and the channel dimension of the language features. The first token $w_1$ of linguistic expression features is a special [CLS] token, which is the global representation that understands the language expression at the sentence level. Following the previous methods (Yang et al., 2022; Liu et al., 2023a; Tang et al., 2023; Yang et al., 2023), we apply the early fusion at each stage of the vision encoder to align the vision features and the linguistic expression features.

## 3.2 SEMANTIC VISUAL INFORMATION SELECTION FOR VISUAL EXPRESSION

To improve the guidance capability, we produce the visual expression that contains the visual semantic contexts related to the target object. As illustrated in Figure 2, the visual information selection module consists of the selection stage and the visual expression extraction stage. The selection stage leverages the global-local linguistic cues to capture the rich contextual information. First, the vision features $F_v \in \mathbb{R}^{N \times D}$ and the global-local linguistic tokens $Q_l$ are embedded into the joint embedding space by the linear projection $\phi$, and then the similarity score $S$ of the vision pixel tokens and the linguistic tokens is computed to rank, as follows:

$$X = \phi_{(v)}(F_v) \,, \; Y = \phi_{(l)}(Q_l) \,, \; S = \mathcal{S}(X, Y) \in \mathbb{R}^{T \times N} \,, \; S_{topk} = \mathcal{K}(S) \in \mathbb{R}^{T \times K} \,, \quad (1)$$

where $\mathcal{S}$, $K$, $S_{topk}$ and $\mathcal{K}$ denote the cosine similarity function, the the number of selected pixels, the index list of the selected pixels, and the top-$k$ operation. As shown in Figure 3, the top-$k$ ranked pixel tokens $F_{topk} \in \mathbb{R}^{K \times D}$ for the similarity to each linguistic context are used to produce the visual expression $Q_v = [v_1, v_2, ..., v_T] \in \mathbb{R}^{T \times D}$. Top-$k$ selection enables visual expression tokens to flexibly gather the useful visual information related to the target object to robustly segment the accurate regions. The visual expression tokens capture the semantic visual information from the top-$k$ ranked pixels through the cross-attention mechanism. The process is formulated as follows:

$$m_n = \begin{cases} 1 & if \; n \in S_{topk} \\ 0 & if \; n \notin S_{topk} \end{cases}, \; \widehat{m}_n = \begin{cases} 0 & if \; n \in S_{topk} \\ -\infty & if \; n \notin S_{topk} \end{cases} for \; n \in \{1, 2, ..., N\} \,, \quad (2)$$

$$M_j = [m_1, ..., m_N] \,, \; M = [M_1, ..., M_T] \,, \; \widehat{M}_j = [\widehat{m}_1, ..., \widehat{m}_N] \,, \; \widehat{M} = [\widehat{M}_1, ..., \widehat{M}_T], \quad (3)$$

$$S_{norm} = \texttt{softmax}(S + \widehat{M}), \; E = \frac{1}{K} \sum_{K} (M \cdot S_{norm} \cdot F_v), \; Q_v = \texttt{CrossAtt}(E, F_{topk}), \quad (4)$$

where $M \in \mathbb{R}^{T \times N}$ is the binary mask for the top-$k$ selection, and $E \in \mathbb{R}^{T \times D}$ is the intermediate visual tokens obtained by the top-$k$ weighted average pooling. $\texttt{CrossAtt}(q, kv)$ indicates

the cross-attention layer using $q$ as the query and $kv$ as the key-value. Since the top-$k$ selection is discrete, the normalized top-$k$ ranked similarity score map $S_{norm}$ is obtained by normalizing the whole similarity score map $S$ combined with $\widehat{M}$ that masks the non top-$k$ ranked similarity scores. After producing the visual expression $Q_v$, the visual expression tokens perform the self-attention to mutually complement each visual token's information and capture the visual contextual information, improving the ability to understand the context of the target region (see Appendix A for the pseudocode). The similarity score map $\mathbf{s} \in \mathbb{R}^{1 \times N}$ for the global visual token $v_1$ is supervised by the pixel contrastive loss, calculated as:

$$\mathcal{L}_{cont} = \begin{cases} -\log(\sigma(\mathbf{s}_q/\tau)) & if \ q \in \mathcal{Z}^+ \\ -\log(1 - \sigma(\mathbf{s}_q/\tau)) & if \ q \in \mathcal{Z}^- \end{cases}, \tag{5}$$

where $\mathcal{Z}^+$ and $\mathcal{Z}^-$ denote the set of the relevant pixels and irrelevant pixels for the target regions. $\tau$ is a learnable temperature parameter, and $\sigma$ is a sigmoid function. The pixel contrastive loss encourages that the relevant pixels are embedded closer together for high similarity score and the irrelevant pixels are embedded far apart for low similarity score.

## 3.3 DECODER REFERRING TO VISUAL AND LINGUISTIC GUIDANCE EXPRESSIONS

To segment the target region, the decoder leverages the guidance set $\mathcal{G} = \{Q_l, Q_v\}$ composed of the linguistic expression tokens and visual expression tokens, which are produced to enrich the target object information. At each decoder stage, the cross-attention layer that uses the vision features as the query and the guidance set as the key-value is employed to highlight the target region by referring to the guidance set. After that, the vision decoder features are upsampled by the bilinear interpolation, and concatenated with the corresponding vision encoder features. The concatenated features are fed into the next decoder stage. The final segmentation map is projected to a binary class mask by a linear projection layer. The binary cross-entropy loss is used for the network training.

## 4 EXPERIMENTS

### 4.1 IMPLEMENTATION DETAILS

**Experimental settings.** Our method was implemented in PyTorch (Paszke et al., 2019). The vision encoder is Swin-B (Liu et al., 2021) initialized with the pre-trained weight on ImageNet-22K (Krizhevsky et al., 2012), and the language encoder is BERT-base (Devlin et al., 2018) initialized with using the official pre-trained weight of uncased version. The decoder was randomly initialized. We trained the model for 40 epochs with a batch size of 16 on 4 RTX 3090 GPUs. We used the AdamW optimizer with initial learning rate of 3e-5 and adopted the polynomial learning rate decay scheduler. The input image resolution was $480 \times 480$ pixels. The maximum sequence length was set to 21 words including the [CLS] token for all datasets. All ablations were conducted on RefCOCO.

**Datasets.** RefCOCO (Yu et al., 2016) and RefCOCO+ (Yu et al., 2016) are widely utilized datasets collected from the MSCOCO dataset for referring image segmentation. RefCOCO contains 19,994 image with 142,209 language expressions for 50,00 object, and RefCOCO+ contains 19,992 images with 141,564 expressions for 49,856 objects. Each expression in RefCOCO and RefCOCO+ consists 3.5 words on average. RefCOCO and RefCOCO+ contains 3.9 object of the same category per image on average. The expression in RefCOCO+ do not include words about absolute locations, which makes it more challenging than RefCOCO. G-Ref (Mao et al., 2016) is another commonly used dataset, which was collected from Amazon Mechanical Turk. G-Ref contains 26,711 images with 104,560 expressions for 54,822 objects. Compared with RefCOCO and RefCOCO+, G-Ref has more complex expressions by consisting longer sentence length containing 8.4 words on average. Thus, it is a more challenging dataset than RefCOCO and RefCOCO+.

**Evaluation metrics.** Following the previous works (Yang et al., 2022; Ding et al., 2022; Wang et al., 2022), we adopted the overall intersection-over-union (oIoU), mean intersection-over-union (mIoU), and precision at 0.5, 0.7 and 0.9 thresholds. The oIoU is the ratio between the total intersection regions and the total union regions of all test samples. The mIoU is the average of IoUs between the predicted mask and the ground truth of all test samples. The precision is the percentage of test samples that have an IoU score higher than the threshold.

| Method | Backbone | RefCOCO | | | RefCOCO+ | | | G-Ref | | |
|--------|----------|---------|--------|--------|----------|--------|--------|---------------|-----------------|---------------|
| | | val | test A | test B | val | test A | test B | val$_{(U)}$ | test$_{(U)}$ | val$_{(G)}$ |
| RRN (Li et al., 2018) | ResNet101 | 55.33 | 57.26 | 53.93 | 39.75 | 42.15 | 36.11 | - | - | 36.45 |
| MAttNet (Yu et al., 2018) | ResNet101 | 56.51 | 62.37 | 51.70 | 46.67 | 52.39 | 40.08 | 47.64 | 48.61 | - |
| BRINet (Hu et al., 2020) | ResNet101 | 60.98 | 62.99 | 59.21 | 48.17 | 52.32 | 42.11 | - | - | 47.57 |
| CMPC (Huang et al., 2020) | ResNet101 | 61.36 | 64.53 | 59.64 | 49.56 | 53.44 | 43.23 | - | - | 49.05 |
| MCN (Luo et al., 2020) | DarkNet53 | 62.44 | 64.20 | 59.71 | 50.62 | 54.99 | 44.69 | 49.22 | 49.40 | - |
| EFN (Feng et al., 2021) | ResNet101 | 62.76 | 65.69 | 59.67 | 51.50 | 55.24 | 43.01 | - | - | 51.93 |
| LTS (Jing et al., 2021) | DarkNet53 | 65.43 | 67.76 | 63.08 | 54.21 | 58.32 | 48.02 | 54.40 | 54.25 | - |
| SeqTR (Zhu et al., 2022) | DarkNet53 | 67.26 | 69.79 | 64.12 | 54.14 | 58.93 | 48.19 | 55.67 | 55.64 | - |
| ReSTR (Kim et al., 2022) | ViT-B | 67.22 | 69.30 | 64.45 | 55.78 | 60.44 | 48.27 | 54.48 | - | - |
| CRIS (Wang et al., 2022) | CLIP-R101 | 70.47 | 73.18 | 66.10 | 62.27 | 68.08 | 53.60 | 59.87 | 60.36 | - |
| LAVT (Yang et al., 2022) | Swin-B | 72.73 | 75.82 | 68.79 | 62.14 | 68.38 | 55.10 | 61.24 | 62.09 | - |
| VLT (Ding et al., 2022) | Swin-B | 72.96 | 75.96 | 69.60 | 63.53 | 68.43 | 56.92 | 63.49 | 66.22 | 62.80 |
| RefSegformer (Wu et al., 2022) | Swin-B | 73.22 | 75.64 | 70.09 | 63.50 | 68.69 | 55.44 | 62.56 | 63.07 | 58.48 |
| ReLA (Liu et al., 2023a) | Swin-B | 73.82 | 76.48 | 70.18 | 66.04 | 71.02 | 57.65 | 65.00 | 65.97 | 62.70 |
| DMMI (Hu et al., 2023) | Swin-B | 74.13 | 77.13 | 70.16 | 63.98 | 69.73 | 57.03 | 63.46 | 64.19 | 61.98 |
| SADLR (Yang et al., 2023) | Swin-B | 74.24 | 76.25 | 70.06 | 64.28 | 69.09 | 55.19 | 63.60 | 63.56 | 61.16 |
| CGFormer (Tang et al., 2023) | Swin-B | 74.75 | 77.30 | 70.64 | 64.54 | 71.00 | 57.14 | 64.68 | 65.09 | 62.51 |
| **VEGA (Ours)** | Swin-B | **75.35** | **77.97** | **71.94** | **66.70** | **72.08** | **59.85** | **65.78** | **66.93** | **63.49** |

Table 1: Performance comparison with previous state-of-the-art methods using oIoU (%) on three public RES benchmarks. The best results are in **bold**. U: UMD split. G: Google split.

| Guidance Set Element | P@0.5 | P@0.7 | P@0.9 | mIoU (%) | oIoU (%) |
|----------------------|-------|-------|-------|----------|----------|
| linguistic expression | 85.47 | 76.27 | 35.79 | 75.16 | 73.31 |
| enhanced linguistic expression | 86.39 | 78.32 | 36.89 | 76.06 | 73.91 |
| enhanced linguistic expression + visual expression (**Ours**) | **86.71** | **78.30** | **37.24** | **76.28** | **75.35** |

Table 2: Comparison of our method with two conventional linguistic guidance elements. "linguistic expression": not refer to the visual information. "enhanced linguistic expression": enhanced by the cross-attention using vision features as a key-value.

## 4.2 COMPARISON WITH THE STATE-OF-THE-ART

In Table 1, we evaluated our approach with previous state-of-the-art methods on three public benchmarks for referring expression segmentation. Our method consistently showed strong performance on all evaluation splits of all datasets, and outperformed other existing methods on three benchmarks. Compared to VLT (Ding et al., 2022), which leverages the enhanced linguistic features as the guidance set element, our VEGA improved performance by 2.39%, 2.01% and 2.34% on each split of RefCOCO, respectively. Compared to the recent state-of-the-art method, ReLA (Liu et al., 2023a), for more challenging benchmark RefCOCO+, our model showed 0.66%, 1.06% and 2.20% higher performance on each split. In addition, for the most challenging benchmark G-Ref, our VEGA achieved significant improvements of 0.78%, 0.96% and 0.79% on each split.

## 4.3 ABLATION STUDIES

### 4.3.1 EFFECTIVENESS OF OUR VISUAL EXPRESSION AS A GUIDANCE SET ELEMENT

In Table 2, we conducted experiments to prove the effectiveness of using the visual expression as well as the language expression as the element of our guidance set. Compared to "linguistic expression" method, "enhanced linguistic expression" method showed a 0.60% oIoU improvement. This suggests that the enhancement of the language features by referring to the visual information helps to improve the comprehension for the meaning of the language expression. Compared to these two methods, our "visual expression" method, which flexibly selects the unique visual information relevant to the target object, showed significant improvements of 2.04% and 1.44% oIoU, respectively. These results indicate that "enhanced linguistic expression" method cannot provide the unique visual information sufficiently to the network, and using "visual expression" as a guidance set element is more beneficial for the robust segmentation by effectively enhancing the guidance set capacity.

### 4.3.2 EFFECTIVENESS OF THE PROPOSED VISUAL INFORMATION SELECTION MODULE

In Table 3a, we conducted the ablation study on the effectiveness of the proposed visual information selection module. Our module is composed of the flexible selection stage and the visual expression extraction stage. To evaluate the effectiveness of the selection stage more rigorously, we compared with a non-selection method (row 1) that uses all visual pixels as the visual guidance element. The "total visual pixel tokens" method contains the non-target region information, and the "top-k

| Visual Guidance Element | P@0.5 | P@0.7 | P@0.9 | mIoU (%) | oIoU (%) |
|---|---|---|---|---|---|
| total visual pixel tokens | 86.14 | 77.76 | 36.87 | 76.02 | 74.16 |
| top-$k$ visual pixel tokens | 86.30 | 77.95 | 37.01 | 76.04 | 74.58 |
| visual expression (**Ours**) | **86.71** | **78.30** | **37.24** | **76.28** | **75.35** |

| Global | Local | P@0.5 | P@0.7 | P@0.9 | mIoU (%) | oIoU (%) |
|---|---|---|---|---|---|---|
| ✗ | ✗ | 86.19 | 77.93 | 36.81 | 75.83 | 73.84 |
| ✗ | ✓ | 86.44 | 78.11 | 37.06 | 76.10 | 74.86 |
| ✓ | ✗ | 86.56 | 78.13 | 37.10 | 76.13 | 74.93 |
| ✓ | ✓ | **86.71** | **78.30** | **37.24** | **76.28** | **75.35** |

(a) Ablation with different visual guidance elements.  (b) Effect of the global-local contexts.

Table 3: Ablation studies on the effectiveness of (a) our visual information selection module and (b) considering the global-local contexts to extract the visual expression.

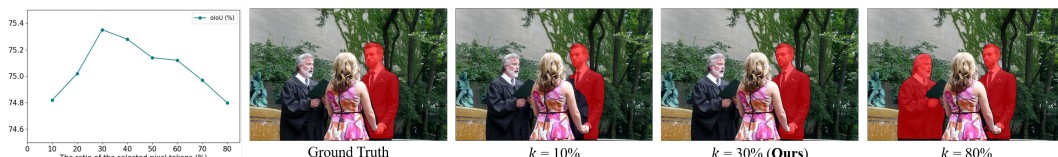

Figure 4: Ablation on number of $k$. Performance by increasing the $k$ (column 1) and visualized results at different $k$.

visual pixel tokens" method is flexibly selected via the top-k selection using the global context cue for guidance to the target region. Even though the non-selection method uses more than three times pixel tokens as a guidance set than the selection method (row 2), the selection method showed 0.42% higher oIoU performance. This indicates that flexibly selecting the useful visual information relevant to the target object is better to robustly guide the target regions than using all of pixels. Furthermore, our "visual expression", which is produced at the visual expression extraction stage, achieved 0.77% oIoU improvement compared to "top-$k$ visual pixel tokens" method. This suggests that considering the contextual information between the visual expression tokens improves the ability to understand the context of the target region.

### 4.3.3 EFFECTIVENESS OF CONSIDERING THE GLOBAL AND LOCAL CONTEXTS

We conducted the ablation on the effectiveness of the global-local linguistic context-aware visual tokens to use the visual information in the guidance set. As shown in Table 3b, removing the use of the local context cues leads to a decrease of 0.42% in oIoU compared to the full model (row 4). In addition, removing the use of the global context cues leads to a 0.49% drop in oIoU, and removing the use of both global and local context cues leads to a significant drop of 1.51% oIoU performance. These results indicate that using global and local context-aware visual tokens as the guidance set elements helps to enrich the visual contextual information to strengthen the precise region.

### 4.3.4 NUMBER OF $k$

We experimented on the optimized value of $k$ (%), which is the ratio of the pixel tokens selected for the visual expression extraction to flexibly collect the useful visual information. As shown in Figure 4, the $k$ of 30 showed 0.46% and 0.48% oIoU improvements compared to $k$ of 10 and 80, respectively (see Appendix C for detailed results). The smaller number of $k$ resulted in a lack of information, where the useful visual information cannot be sufficiently exploited. In contrast, the larger number of $k$ resulted in including the noise information and degrades the guidance capability. Therefore, beyond finding the optimal $k$, these results demonstrate that the proposed selection-based framework improves the capacity for the robust guidance.

### 4.3.5 DESIGN CHOICES

**Selection method.** As described in Eq.1, the similarity score between vision pixels and the linguistic guidance tokens is computed to select the useful visual information relevant to the target regions. To better select the visual information based on the similarity scores, we conducted experiments with the thresholding method and the top-$k$ method in Table 4a. The top-$k$ selection method showed 0.54% higher oIoU performance than the thresholding method. This indicates that the top-$k$ selection can flexibly gather the semantic visual information, whereas the thresholding method cannot sufficiently collect the informative pixel tokens. Thus, our module can enhance the adaptability for the diverse linguistic expressions and image contexts by leveraging the top-$k$ selection.

**Applying language-to-pixel contrastive loss.** In Table 4b, we experimented on applying the language-to-pixel contrastive loss (see Eq.5) to the similarity score map obtained by the global

| Method | mIoU | oIoU |
|---|---|---|
| sigmoid $(> 0.5)$ | 76.15 | 74.81 |
| top-$k$ ranked selection | **76.27** | **75.35** |

(a) **Selection method.**

| Contrastive loss | mIoU | oIoU |
|---|---|---|
| ✗ | 76.24 | 74.92 |
| ✓ | **76.27** | **75.35** |

(b) **Language-to-pixel contrastive loss.**

| Mul. similarity scores | mIoU | oIoU |
|---|---|---|
| ✗ | 76.23 | 75.05 |
| ✓ | **76.27** | **75.35** |

(c) **Multiply similarities by top-$k$ pixels.**

| Normalization | top-$k$ mask | mIoU | oIoU |
|---|---|---|---|
| ✗ | w/o | 76.18 | 74.97 |
| ✓ | w/o | 76.22 | 75.14 |
| ✓ | w/ | **76.27** | **75.35** |

(d) **Softmax normalization with top-$k$ mask.**

| top-$k$ cross-att. | mIoU | oIoU |
|---|---|---|
| ✗ | 76.11 | 74.23 |
| ✓ | **76.27** | **75.35** |

(e) **Visual expression token extraction.**

Table 4: Ablation experiments on the design of our visual information selection module. Our default settings are marked in gray .

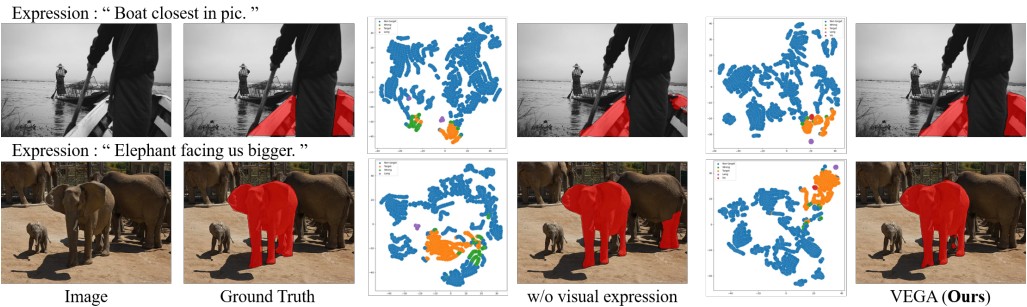

Figure 5: Comparison of t-SNE results with an ablated model. Orange: target pixel. Blue: non-target pixel. Green: wrong predicted pixel. Purple: linguistic expression. Red: visual expression.

context token. This result indicates that the language-to-pixel contrastive loss helps to supervise the selection of semantic pixel tokens associated with the global context token.

**Others.** (1) We multiplied the different weights based on the similarity score by the top-$k$ pixel tokens before extracting the visual expression tokens (see Eq.4). In Table 4c, this showed better performance than non-multiplication method. (2) We also ablated on applying a softmax normalization to the top-$k$ similarity scores before the multiplication. In Table 4d, the normalization of the top-$k$ similarity scores achieved higher performance than both the normalization of the total pixels similarity scores and the non-normalization. (3) In Table 4e, we conducted the ablation on performing the cross-attention using the intermediate visual tokens as the query and the top-$k$ pixels as the key-value (see Eq.4) to extract the visual expression. Performing the cross-attention showed an improvement of 1.12% oIoU. This indicates that the cross-attention with top-$k$ pixels enables the visual expression tokens to capture the semantic information from the top-$k$ pixel tokens.

### 4.4 QUALITATIVE RESULTS

We visualized t-SNE results of our VEGA compared to the ablated model (*i.e.* w/o visual expression) in Figure 5. The t-SNE exhibits the distribution of the pixel tokens and the visual and linguistic expressions. As shown in the t-SNE results, our visual expression tokens are embedded closer to the target pixels, whereas the linguistic expression tokens of the ablated model do not provide sufficient guidance for target pixel tokens. Therefore, these results indicate that our visual expression can complement the linguistic information and lead to the robust segmentation. In Figure 6 (a), we visualized the segmentation results of our VEGA with previous methods for diverse types of inputs. Our VEGA segmented the target regions more clearly for the complex and ambiguous language expressions (*e.g.* (1) and (2)) and the complicated images (*e.g.* (4) and (5)), whereas other methods incorrectly predicted the objects and uncertainly segmented the regions. In Figure 6 (b) and (c), we visualized the results for the challenging types of language expressions to verify the robustness of our method, such as typos (*e.g.*, "seond" and "blu") and slang (*e.g.*, "hitta" and "drk"), which make it difficult for the network to refer to the linguistic contexts. Compared previous methods, our VEGA correctly determined the target regions. In Figure 7, we visualized the results for different language expressions describing the same target. Our method consistently predicted the accurate regions by leveraging the visual expression, which complements the guidance information, with the language expression, while the ablated model showed inconsistent predictions and segmented the non-target regions. These qualitative results demonstrate that our method enhances the robustness

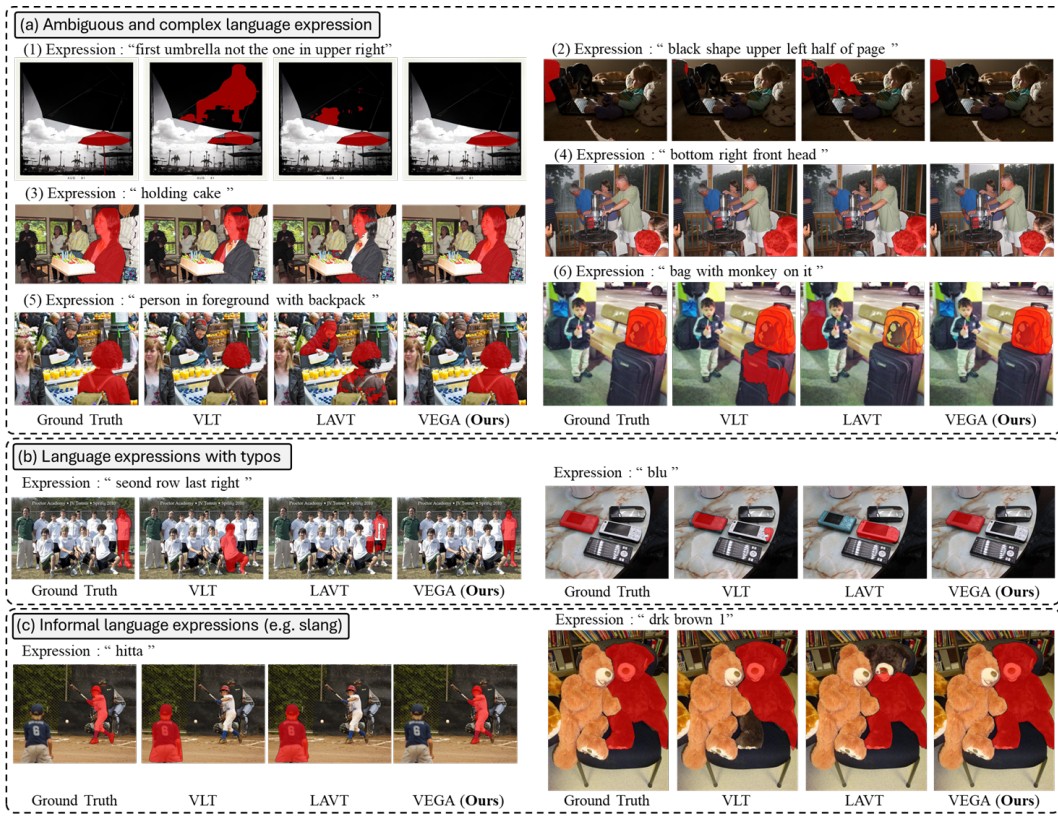

Figure 6: Visualization of our method and the previous method on (a) different types of images and language expressions and (b-c) challenging types of linguistic expressions.

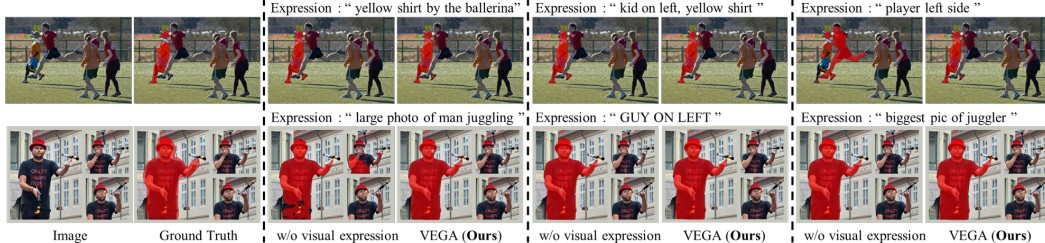

Figure 7: Visualization of our method and w/o visual expression model on various language expressions describing the same target object in the image.

of the guidance capacity and improves the ability to comprehend the context of the target region. We provide additional qualitative results in Appendix D.

## 5 CONCLUSION

We propose a novel visual expression guidance framework for referring expression segmentation, VEGA, which enables the network to refer to the visual expression that complements the linguistic expression information by providing the useful visual information relevant to the target regions. Our framework enhances the guidance capacity to robustly segment the accurate regions. To better produce the visual expression, we introduce a novel visual information selection module that flexibly selects the semantic visual information related to the target regions, enhancing the adaptability to various language expressions and image contexts. This module also allows each token of the visual expression to consider the visual contextual information by exploiting the global-local linguistic cues, improving the ability to understand the context of the target regions. Experiments demonstrate the effectiveness of our method on three public benchmarks for referring expression segmentation.

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

APPENDIX

- Code and README file were submitted as a zip file for reproducibility.
- In Appendix A, we provide the pseudocode of our visual information selection module.
- In Appendix B, we provide detailed quantitative results of the proposed method on Ref-COCO, RefCOCO+ and G-Ref benchmarks.
- In Appendix C, we present the detailed performance for ablation on number of $k$.
- In Appendix D, we provide additional qualitative comparisons on the various types of language expressions and the different language expressions describing the same target object.

## A    PSEUDOCODE OF OUR VISUAL INFORMATION SELECTION MODULE

**Algorithm 1:**  Visual Information Selection module: PyTorch-like pseudocode

```
# F_v:  vision features, Q_l:  linguistic expression features
# mask_l:  attention mask, matmul:  matrix multiplication
# Cross-Attention(q, k, mask), Self-Attention(q, k, mask)

def VISBlock(F_v, Q_l, mask_l, ratio=0.3):
    # joint multimodal embedding
    X = l2norm(proj(F_v), axis=-1) # [B, N, D]
    Y = l2norm(proj(Q_l), axis=-1) # [B, T, D]

    # calculate cosine similarities
    S = matmul(Y, X.T) # [B, T, N]

    # top-k ranking of similarity scores
    rank_index = topk(S, k=int(N*ratio), dim=-1) # [B, T, K]

    # top-k mask generation
    M = binary_mask(rank_index) # [B, T, N]

    # non top-k similarity scores masking
    # & top-k ranked similarities normalization
    M_hat = M * 1e4 - 1e4 # [B, T, N]
    S_norm = softmax((S + M_hat), dim=-1) # [B, T, N]

    # expand dimension of F_v, M, S_norm
    F_v = repeat(F_v.unsqueeze(1), T, dim=1) # [B, T, N, D]
    M = M.unsqueeze(-1) # [B, T, N, 1]
    S_norm = S_norm.unsqueeze(-1) # [B, T, N, 1]

    # weighted average summation
    E = sum((M * S_norm * F_v), dim=2) / int(N*ratio) # [B, T, D]

    # the visual expression extracted by the cross-attention with
     top-k pixels
    Q_v = Cross-Attention(E, F_v, M_hat) # [B, T, D]

    # considering the visual contexts via the self-attention
    Q_v = Self-Attention(Q_v, Q_v, mask_l) # [B, T, D]

    # new guidance set
    G = concat([Q_l, Q_v], dim=1) # [B, 2*T, D]
    return G
```

# B  DETAILED QUANTITATIVE RESULTS

Table 5 showed the detailed precision, mIoU, and oIoU scores of our model on RefCOCO, Ref-COCO+ and G-Ref datasets to complement the quantitative results.

| Dataset | | P@0.5 | P@0.6 | P@0.7 | P@0.8 | P@0.9 | mIoU (%) | oIoU (%) |
|---|---|---|---|---|---|---|---|---|
| RefCOCO | val | 86.70 | 83.69 | 78.28 | 67.11 | 37.24 | 76.27 | 75.35 |
| | test A | 89.00 | 86.58 | 82.09 | 70.48 | 37.05 | 78.02 | 77.97 |
| | test B | 82.41 | 79.00 | 73.17 | 63.14 | 37.45 | 73.02 | 71.94 |
| RefCOCO+ | val | 76.75 | 73.63 | 68.64 | 59.17 | 32.48 | 67.74 | 66.70 |
| | test A | 83.09 | 80.65 | 75.48 | 65.25 | 34.19 | 72.81 | 72.08 |
| | test B | 67.78 | 64.45 | 58.83 | 49.68 | 29.37 | 60.82 | 59.85 |
| G-Ref | val$_{(U)}$ | 75.94 | 70.98 | 63.99 | 51.90 | 27.29 | 67.22 | 65.78 |
| | test$_{(U)}$ | 76.13 | 71.31 | 64.60 | 53.25 | 27.87 | 67.45 | 66.93 |
| | val$_{(G)}$ | 73.22 | 68.79 | 63.00 | 52.77 | 27.96 | 65.36 | 63.49 |

Table 5: Precision, mean IoU (%) and overall IoU(%) of the proposed method on three public referring expression segmentation benchmark datasets.

# C  DETAILED ABLATION RESULTS ON NUMBER OF $k$

We provide the detailed ablation results on number of $k$. Table 6 showed the detailed precision, mIoU, and oIoU scores of the different $k$.

| Number of $k$ (%) | P@0.5 | P@0.7 | P@0.9 | mIoU (%) | oIoU (%) |
|---|---|---|---|---|---|
| 10 | 86.67 | 78.21 | 36.02 | 76.11 | 74.82 |
| 20 | 86.56 | 78.25 | 37.18 | 76.08 | 75.02 |
| 30 | **86.70** | **78.28** | **37.24** | **76.27** | **75.35** |
| 40 | 86.59 | 78.26 | 37.22 | 76.24 | 75.28 |
| 50 | 86.49 | 78.17 | 37.23 | 76.22 | 75.14 |
| 60 | 86.35 | 78.25 | 37.21 | 76.16 | 75.12 |
| 70 | 86.34 | 78.24 | 37.19 | 76.08 | 74.97 |
| 80 | 86.28 | 78.22 | 37.14 | 76.03 | 74.80 |

Table 6: Ablation study on number of $k$.

# D  VISUALIZATIONS

In Figure 8, we visualized additional qualitative results on various types of the language expressions and the images to clearly demonstrate the high level of competence in understanding the context of the target regions. Our VEGA showed more accurate segmented regions than the previous state-of-the-art methods for the diverse expressions describing the relative location (*e.g.* "animal behind fence" and "banana closest to apples"), color (*e.g.* "white" and "beige") and other attributes (*e.g.*, "200999", "empty" and "with handles"). In Figure 9, we visualized the results for two or three different language expressions describing the same object. Our method showed robust segmentation for various language expressions, whereas the ablated model segmented the non-target regions or did not highlight the target regions.

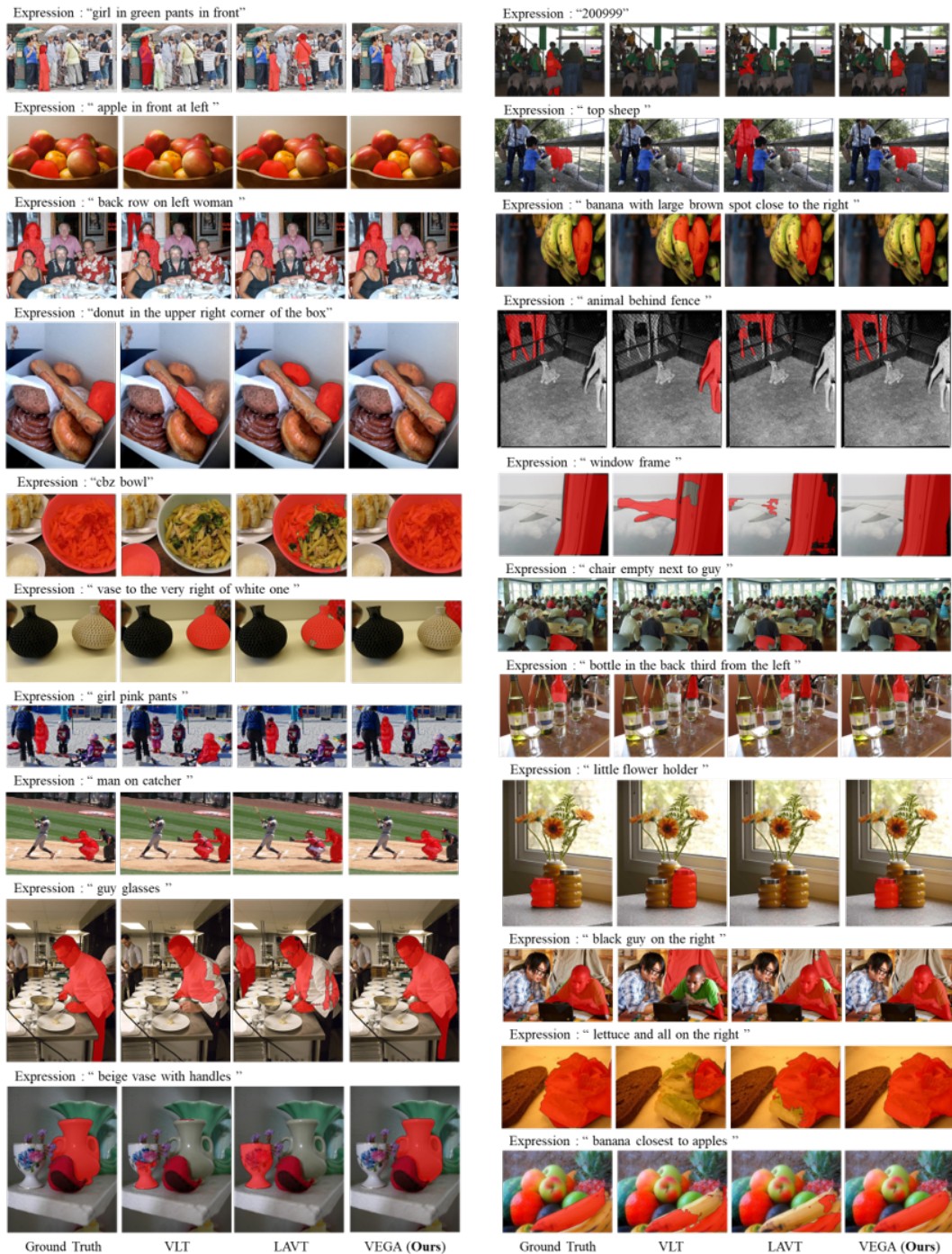

Figure 8: Additional qualitative results of the proposed method and the previous state-of-the-art methods on more diverse language expressions and images.

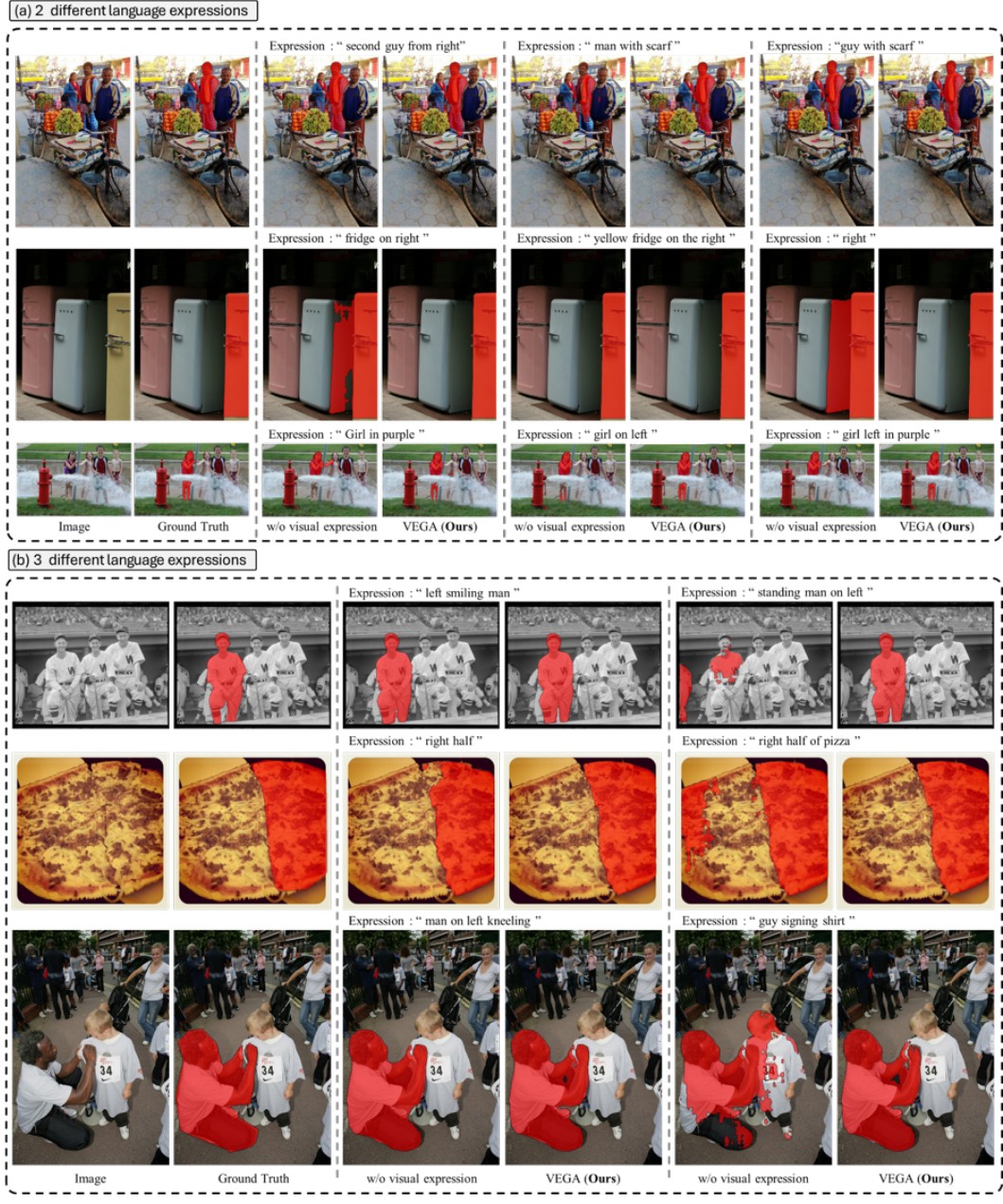

Figure 9: Additional qualitative results of the proposed method and the ablated model on different language expressions describing the same object in the image.

