# OpenReview forum: "VEGA: Visual Expression Guidance for Referring Expression Segmentation"
_ICLR.cc/2024/Conference — ICLR 2024 Conference Withdrawn Submission_

### Official Review · Reviewer_qp2v · 2023-10-30

**Soundness:** 2 fair
**Presentation:** 2 fair
**Contribution:** 1 poor
**Rating:** 3
**Confidence:** 5

**Summary:**

The authors propose a visual expression guidance framework for referring expression segmentation, called VEGA. VEGA enables the network to refer to the visual expression that complements the linguistic expression information by providing relevant visual information to the target regions. A visual information selection module is introduced to select the semantic visual information related to the target regions, enhancing adaptability to various language expressions and image contexts. Experiments show the proposed method obtain good performance.

**Strengths:**

* The proposed method achieves good performance on three referring expression segmentation datasets.

* The paper writing is good and easy to follow.

**Weaknesses:**

* The innovation of the Visual Information Selection module is limited. K-Net [1] has already proposed a method for selecting top-k enhanced visual features in the universal segmentation field. SADLR applies the idea of K-Net to referring expression segmentation tasks, and PPMN [2] also applies top-k selection to enhance phrase features in a similar panoptic narrative grounding task. Therefore, the reviewers believe that this module has minimal differentiation from previous methods.

* The reviewers have some doubts about the implementation of the Visual Information Selection module. In Equation (4), S_norm already sets the similarity of pixel tokens not belonging to the top-k to 0, so why is it necessary to multiply it with M? Additionally, there are two multiplication operations in the middle part of Equation (4) that use the same symbol. If they are both element-wise multiplication or matrix multiplication, it seems incorrect in terms of dimensions. Furthermore, E is already an image feature obtained by weighted summation of K pixels, so what is the purpose of performing cross-attention again?

* From the ablation experiments, the performance improvement from selecting top-k and visual expression seems marginal. The state-of-the-art performance of this paper may be achieved based on a strong baseline, which does not provide strong support for the effectiveness of the proposed innovations.

[1] Zhang et al. K-net: Towards unified image segmentation. NeurIPS 2021

[2] Ding et al. PPMN: Pixel-Phrase Matching Network for One-Stage Panoptic Narrative Grounding. ACM MM 2022.

**Questions:**

See Weaknesses.

---

### Official Review · Reviewer_LTim · 2023-10-30

**Soundness:** 3 good
**Presentation:** 3 good
**Contribution:** 2 fair
**Rating:** 3
**Confidence:** 4

**Summary:**

This paper proposes a Visual Expression GuidAnce framework for referring expression segmentation, which enables the network to refer to the visual expression that complements the linguistic expression information to improve the guidance capability. The proposed semantic visual information selection leverages the similarity between word tokens and pixel tokens to select top-$k$ pixel tokens for each word token, which are used to collect the semantic information relevant to the target regions by cross-attention mechanism. Extensive experimental results on three benchmark datasets show the effectiveness of the proposed method.

**Strengths:**

The proposed method is well-motivated and technically sound. The paper is well-organized and shows state-of-the-art performance. The qualitative results are also adequate the visually show its effectiveness.

**Weaknesses:**

1. The proposed method is with limited novelty. The visual information selection module, which includes the top-k selection and visual expression extraction, is simple and straightforward.
2. The performance gain over existing methods is marginal. Besides, according to Table 2, the proposed visual expression is of limited effect.

**Questions:**

What is the limitation of this method? Please show some failure cases of this method.

---

### Official Review · Reviewer_PVNu · 2023-11-04

**Soundness:** 2 fair
**Presentation:** 3 good
**Contribution:** 2 fair
**Rating:** 3
**Confidence:** 5

**Summary:**

The paper proposes a new framework to tackle referring image segmentation. In contrast to other works that only use text tokens to segment the target object, the proposed framework makes use of both visual tokens and text tokens to guide the segmentation. To do so, they develop a selection module that first gets top-k image features based on their similarity with text tokens, then goes through a set of transformer layers to obtain the visual tokens. Experiments on several datasets show that the proposed method is robust and effective.

**Strengths:**

1. This paper is well-written and easy to follow.
2. Experimental results show improvements on 3 datasets.

**Weaknesses:**

1. The motivation is confusing. Why do we need to use visual knowledge to complete text? If the text query is enough to localize the target object, it is not necessary to complete it. If not, how can we know the target object and complete the text?
2. What will happen if noisy complements are generated?
3. Many previous works also incorporate vision knowledge into text, such as [A-C]. What is the difference between the proposed method with them?

[A] Key-word-aware network for referring expression image segmentation. In ECCV, 2018.
[B] Cross-modal self-attention network for referring image segmentation, In CVPR 2019.
[C] See-through-text grouping for referring image segmentation. In ICCV, 2019.

**Questions:**

Please see weakness.